# Decision analysis of PPP project's parties based on deep consumer participation

**Wei Liu** *, **Xiaoli Wang, Sheng Jiang**

School of economics and management, Tongji University, Shanghai, China

* leuvei@163.com

## Abstract

Although PPP(Public-private partnership) mode has been applied for a long time in infrastructural project, the success rate is not very high. The sustainability of PPP projects is still influenced by many factors. In order to examine the evolutionary stable strategies (ESSs) of social capital, government, and paying consumers, a tripartite evolutionary game model is established in this work. In order to further promote consumer participation, it is necessary to make the assumption that customer oversight and review can have an impact on service prices. The results show: i)The strategy choice of consumer depends on the comparison between supervision cost of consumer and price coefficient for consumer to social capital. ii) Consumer supervision can promote the provision of high-quality services by social capital. iii)The difference between high-quality cost and low-quality cost, subsidy coefficient, price coefficient and supervision cost of consumer are critical factors influencing both evolutionary results and trajectories. This paper also puts forward policy implications for the three stakeholders to promote social capital's high-quality strategy so as to maintain the sustainability of PPP projects.

## 1. Introduction

A large number of quasi-public and public services are needed to be provided by the government. If these projects are funded entirely through the government, the government will face a huge fiscal gap. Public-private partnership (PPP) has been widely adopted by governments and is considered a beneficial manner for promoting public infrastructure development around the world [1]. The World Bank defines PPP as a long-term cooperative relationship established between the government and the private sector through the joint promotion of the construction of public infrastructure. The private sector is responsible for the construction and operation, taking risks and corresponding responsibilities, and the government is responsible for the regulation [2]. First of all, the PPP model can solve the project's capital needs through social financing. The government can introduce the professional management of social capital through the PPP mode to improve the efficiency of project management and operation. A successful PPP project requires a healthy partnership between social capital and the government. social capital's investment needs to be given a reasonable income. Governments need to ensure the public's interest.

**Data Availability Statement:** All relevant data are within the manuscript and its Supporting information files.

**Funding:** The author(s) received no specific funding for this work.

**Competing interests:** The authors have declared that no competing interests exist.

Unlike public services that are funded by the government, projects like cultural tourism, industrial parks, and aged care are ultimately funded by customers. Consumers have the right to take part in supervision as service recipients. Customers are more willing than producers or employees to supervise goods or services [3]. The administration views public governance as requiring greater public involvement in the long-term success of the programs. Among the PPP projects in China, By 2021, the total number of PPP projects was 10,472, of which culture, education, tourism, health, and elderly care projects, mainly paid by consumers, accounted for only 13.9 percent [4]. The PPP projects of the user-pay model have great development potential in China.

Some academic articles investigate the causes of PPP project failure. Through examining 16 failed water PPPs, Tariq [5] revealed a number of potential failure causes, such as the public sector's incapacity, the private sector's subpar technical performance, macroeconomic instability, conflicts between the public and private sectors, social and political opposition to privatization, and a lack of transparency. Soomro [6] discovered that some of the reasons for 35 failed transportation PPPs globally include a lack of client trust, unproductive business procedures, and subpar work produced by concessionaires. Consumer opposition to public-private partnership (PPP) transportation projects could be caused by political influence favoring specific interest groups and high toll fees [7]. Poor customer service, service interruptions, and erroneous bills given to consumers are the most important issues, while public demonstrations against privatization are the ones that happen the most frequently, according to Tariq [8]. Due to the existence of these factors, the operation of the project requires scientific management and explicit regulation [9]. The risk of market failures can be decreased and the interests of all parties concerned can be safeguarded by reasonable regulation [10].

There are some studies that identify factors affecting the operation of the PPP projects. Using exploratory factor analysis, Li [11] identifies 27 factors that affect the sustainability of water environment treatment PPP projects across five dimensions: economy, society, resources and environment, engineering, and project management. Khahro [12] presents the identified key risks of PPP projects in developing countries, which are mostly financial and public-oriented. It is concluded that inflation, revenue risk from end-users, foreign exchange fluctuation, political situation, law and order, and corruption are the major risks in developing countries for better management of PPP projects. Soojin Kim [13] offers for policymakers that better risk allocation and proper, mutual coordination between the public and private partners represent essential factors for PPP success.

The overarching contribution of Ivan [14] is to explore the stakeholders' perceptions of PPP policy in universal secondary education (USE) and its CSFs in Uganda. Babatund [15] identified and assessed the perceptions of stakeholders on critical success factors (CSFs) for public-private partnership (PPP) projects in Nigeria. Li [8] thinks that public participation has an impact on the project's profit and public and social capital behavior. The results of a survey of PPP projects in the UAE show that proper risk allocation and sharing among project stakeholders is one of the five most important key success factors. Toriola-Coker [16] believes that the main factor hindering the development of PPP projects in developing countries is the marginalization of end consumers in PPP projects. Active participation of end user stakeholders from start to finish will address the marginalization of end user stakeholders in the PPP road project [17].

Researchers have studied how citizens can participate in public governance. Focusing on the governance facets of China's environmental policymaking and the prerequisites for meaningful public engagement in sustainable urbanization policymaking are the objectives of Enserink [18]. According to Anne N. Glucker [19], public participation can promote democratic policy-making and increase the efficacy of environmental impact assessments. By

improving crucial elements like stakeholder engagement, user fees, risk allocation regulations, and so forth, Nilesh and Boeing [20] suggest opportunities to improve sustainable growth in the PPP operating process. Li [21] believes that in order to control the actions of the private sector and boost the effectiveness of supervision, the government should increase public influence and engagement through the use of the Internet, big data, and other cutting-edge technology. Zhang [2] suggests that in the future, researchers and practitioners should focus on finding a balance between government overregulation and deregulation while appeasing the three parties involved in various phases and fields. Consumer behavior study in tripartite evolutionary games in PPP mode in user pay groups is still lacking.

Customers may participate in the processing of tripartite games in PPP projects with designated consumers. Social capital serves the needs of customers by being incorporated into the PPP project. Social capital determines the caliber of services. The government department works to maximize its performance while running PPP projects in order to meet public demands, enhance social welfare, and enhance government reputation. Social capital seeks to increase corporate profits. Liu [22], analyzing the tripartite game in PPP mode in specified consumer groups, constructed the feedback mechanism between consumers and the government. This paper studies evolutionary game analysis under the feedback mechanism between consumers and social capital.

To examine the behavior of the tripartite decision-making process, we shall apply evolutionary game theory. Evolutionary models assume that people choose their strategies through a trial-and-error learning process in which they gradually learn that some strategies work better than others, in contrast to traditional game theory models, which assume that all players are completely rational and have complete knowledge of the details of the game [23]. Numerous studies have used evolutionary game theory in a variety of contexts, including stakeholder decision-making, multi-participant cooperation, supply chain management, and public policy. Machiavellianism and other behavioral strategies can be thought of in terms of evolutionary game theory, which also identifies a huge number of particular hypotheses that have not yet been investigated by personality and social psychologists [24]. To characterize the long-term dynamic process of multi-player game playing in coal mine safety regulation under the assumption of constrained rationality, Liu [25] investigates the application of evolutionary game theory. According to Tian [26], the evolutionary game approach may be used to simulate the development of dynamic attacking methods in a vehicular network, and the results of the simulation can be utilized to measure the reputation management scheme's success in providing protection. Song [27] offers an evolutionary game theory-based application model for the evaluation and analysis of chemical production safety management. Du's [28] goal was to look into the C&D waste management stakeholders' decision-making processes. To describe the acquisition of indirect trust information between nodes and to put forth the associated penalty mechanism to promote cooperative behaviors, Li [29] offers a new trust management mechanism based on game theory. Additionally, it has been applied to research moral hazard under asymmetric knowledge in the construction industry [30], the supply chain government subsidy mechanism [31], and the incentive mechanism for green retrofits [32].

Based on the literature review above, two literature gaps are identified. (i)There have been several studies on PPP contracts, operations, dangers, and funding, but only a small number of them have had participation from pertinent parties. There is, however, a dearth of research on the effects of customer involvement in service quality assessment on service prices in PPP projects. In this article, the service pricing feedback system for consumer evaluation is discussed. (ii)This paper will analyze the effects of customer involvement in service quality supervision in PPP projects on the decisions of social capital and government, finally deciding the game

equilibrium state, using a tripartite evolutionary game model. Using MATLAB, the evolution of ESS points is displayed.

The structure of this paper is as follows: In Section 2, we show the basic assumptions and set up the evolutionary game model of tripartite sides under the PPP project. In Section 3, we analyze the evolutionary stability and find out The equilibrium points of the evolutionary game. In Section 4, we employ MATLAB to simulate the evolutionary path of the seven ESS points mentioned and analyze the key parameters affecting the tripartite decision. In Section 5, we propose the conclusion, policy implications, and future research direction.

## 2. Methodology

### 2.1. Basic assumptions

**Assumption 1.** In the evolutionary game of a PPP project with consumer participation, three parties, namely social capital, government, and consumer, are involved. They all conform to the hypothesis of bounded rationality, learn from each other, imitate each other, and evolve continuously in the process of game play to pursue the maximization of their own interests [32].

**Assumption 2.** Social capital has two strategies: providing high-quality service or low-quality service. The probability of Social capital choosing a high-quality service strategy is $x$($x \in [0, 1]$), while the probability of Japan choosing a low-quality service strategy is $1-x$. The government also has two strategies: regulation and non-regulation. The probability of the government choosing the regulation strategy is set as $y$ ($y \in [0, 1]$), while the probability of the government choosing the non-regulation strategy is $1-y$. There are also two strategies for consumers: supervision and non-supervision. The probability of the consumer choosing a supervision strategy is assumed to be $z$($z \in [0, 1]$) while the probability of its non-supervision strategy is $1-z$.

**Assumption 3.** When the consumer chooses the supervision strategy, it will pay the supervision cost, which is relatively small but should be considered. We assume that the behavior of consumer involvement is an evaluation of the quality of services provided by social capital. Consumers can adjust service prices based on the results of quality evaluations. As we have increased the bargaining power of consumers' participation, when social capital provides high-quality services, consumers' evaluation of service quality will not change the initial service price, and it will not affect the social capital's return R. When social capital provides low-quality services, the consumer's evaluation of service quality will be negative, and the consumer has the right to adjust the service price by a coefficient b ($b \in [0, 1]$). The social capital's return is b*R, and the consumer can obtain (1-b) R. Consumers' supervisory rights are not affected by whether the government participates in regulation. When the government participates in regulation, social capital providing high-quality services will receive government subsidies. When social capital provides low-quality services, the government subsidy will be multiplied by a coefficient named a, and social capital will receive the returns a*V, and the government will receive additional earnings (1-a)V. When the government does not participate in regulation, social capital will receive subsidy V regardless of whether it provides high-quality or low-quality services. However, if social capital provides low-quality services, the government's reputation will be damaged.

**Assumption 4.** When the consumer chooses the non-supervision strategy, it will not attain the additional benefit or loss. Consumer benefits are only the satisfaction of the quality of services provided by the social capital, represented by and. represents consumers' satisfaction with high-quality services, and represents consumers' satisfaction with low-quality services. We assume that is always greater than b1, and Ch is always greater than CL, to ensure that the

**Table 1. Parameters and their definition.**

| Variable | Definition | Note |
|---|---|---|
| R | Revenue of social capital to provide service | $R > 0$ |
| $C_H$ | Cost of social capital to provide high quality service | $C_H > 0$ |
| $C_L$ | Cost of sotial capital to provide low quality servlce | $C_L > 0$ |
| $R_g$ | Regulation benefit of government | $Rg > 0$ |
| $C_g$ | Regulation cost of government | $C_g > 0$ |
| $C_s$ | Supervision cost of consumer | $C_s > 0$ |
| $b_1$ | Satisfaction coefficient for consumer to high quality Service | $b_1 > 0$ |
| $b_2$ | Satlsfactlon coefficlent for consumer to low quallty service | $b_2 > 0$ |
| F | Loss for government adopting non-reguletion strategy with soclal capltal's low-quallty servlce strategy | $F > 0$ |
| a | Subsidy coefficient for government to social capital | $0 \leq a \leq 1$ |
| b | price coefficient for consumer to social capital | $0 \leq b \leq 1$ |
| x | Probability of social capital choosing high-quality service strategy | $0 < x < 1$ |
| y | Probabllty of government choosing regulation strategy | $0 \leq y \leq 1$ |
| z | Probability of consumer choosing supervision strategy | $0 \leq z \leq 1$ |

level of high-quality services is better than the level of low-quality services. Consumers have no bargaining power to adjust service prices. Whether the government participates in regulation, the government's revenue is exactly the same as in assumption 3.

All the notations and their meanings are demonstrated in Table 1.

## 2.2. Evolutionary game model

The payoff matrix of social capital, government, and consumer is shown in Table 2.

Assuming that the expected utility of social capital's high-quality service strategy is $U_{11}$, the expected utility of social capital's low-quality service strategy is $U_{12}$, and the average expected

**Table 2. Tripartite evolutionary game payoff matrix.**

| | strategies | Government | | strategies | |
|---|---|---|---|---|---|
| | | **Regulation(y)** | **Non-reguluation(1-y)** | | |
| **Social capital** | High-quality service(x) | $R - C_H + V$,<br>$R_g - C_g$,<br>$b_1 - C_s$ | $R - C_H + V$,<br>$0$,<br>$b_1 - C_s$ | Supervision(z) | **Consumer** |
| | Low-quality service(1-x) | $R * b - C_L + V * a$,<br>$R_g - C_g + V * (1-a)$,<br>$b_2 - C_s + R * (1-b)$ | $R * b - C_L + V$,<br>$-F$,<br>$b_2 - C_s + R * (1-b)$ | | |
| | High-quality service(x) | $R - C_H + V$,<br>$R_g - C_g$,<br>$b_1$ | $R - C_H + V$,<br>$0$,<br>$b_1$ | Non-supervision(1-z) | |
| | Low-quality service(1-x) | $R - C_L + V * a$,<br>$R_g - C_g + V * (1-a)$,<br>$b_2$ | $R - C_L + V$,<br>$-F$,<br>$b_2$ | | |

utility is $\overline{U}_1$, then:

$$U_{11} = yz(R - C_H + V) + y(1 - z)(R - C_H + V) +$$
$$(1 - y)z(R - C_H + V) + (1 - y)(1 - z)(R - C_H + V) \tag{1}$$

$$U_{12} = yz(bR - C_H + Va) + y(1 - z)(R - C_H + V) +$$
$$(1 - y)z(R - C_H + V) + (1 - y)(1 - z)(R - C_H + V) \tag{2}$$

The average expected payoff of social capital is given by $\overline{U}_1$

$$\overline{U}_1 = xU_{11} + (1 - x)U_{12} \tag{3}$$

The replicated dynamic equation of social capital F($x$) is:

$$F(x) = \frac{dx}{dt} = x(U_{11} - \overline{U}_1) = x(1 - x)(U_{11} - U_{12})$$
$$= x(x - 1)(C_H - C_L - zR - yV + bzR + ayV) \tag{4}$$

Assuming that the expected utility of government's regulation strategy is $U_{21}$, the expected utility of government's non-regulation strategy is $U_{22}$, and the average expected utility is $\overline{U}_2$, then:

$$U_{21} = xz\left(R_g - C_g\right) + x(1 - z)\left(R_g - C_g\right) + (1 - x)z\left(R_g - C_g + (1 - a)V\right)$$
$$+(1 - x)(1 - z)\left(R_g - C_g + (1 - a)V\right) \tag{5}$$

$$U_{22} = xz \cdot 0 + x(1 - z) \cdot 0 + (1 - x)z(-F) + (1 - x)(1 - z)(-F) \tag{6}$$

$$\overline{U}_2 = yU_{21} + (1 - y)U_{22} \tag{7}$$

The replicated dynamic equation of government F($y$) is:

$$F(y) = \frac{dy}{dt} = y(U_{21} - \overline{U}_2) = y(1 - y)(U_{21} - U_{22}) = y(y - 1)$$
$$(C_g - R_g - F - V + aV + xF + xV - axV) \tag{8}$$

Assuming that the expected utility of consumer's supervision strategy is $U_{31}$, the expected utility of government's non-regulation strategy is $U_{32}$, and the average expected utility is $\overline{U}_3$, then:

$$U_{31} = xy(b_1 - C_s) + x(1 - y)(b_1 - C_s) + (1 - x)y(b_2 - C_s + (1 - b)R)$$
$$+(1 - x)(1 - y)(b_2 - C_s + (1 - b)R) \tag{9}$$

$$U_{32} = xyb_1 + x(1 - y)b_1 + (1 - x)yb_2 + (1 - x)(1 - y)b_2 \tag{10}$$

$$\overline{U}_3 = zU_{31} + (1 - z)U_{32} \tag{11}$$

The replicated dynamic equation of consumer F($z$) is:

$$F(z) = \frac{dz}{dt} = z\big(U_{31} - \overline{U}_3\big) = z(1 - z)(U_{31} - U_{32})$$
$$= z(z - 1)(C_s - R + bR + xR - bxR) \tag{12}$$

Therefore, the tripartite evolutionary game process of PPP projects can be expressed by a system of differential equations composed of Eqs (4), (7) and (11):

$$\begin{cases} F(x) = \frac{dx}{dt} = x(x - 1)(C_H - C_L - zR - yV + bzR + ayV) \\ F(y) = \frac{dy}{dt} = y(y - 1)\big(C_g - R_g - F - V + aV + xF + xV - axV\big) \\ F(z) = \frac{dz}{dt} = z(z - 1)(C_s - R + bR + xR - bxR) \end{cases} \tag{13}$$

By analyzing the equation, the stable point of evolutionary game can be obtained:(0,0,1), (0,1,1), (1,1,1), (1,1,0), (1,0,0), (1,0,1), (0,0,0), (0,1,0) and (F($x^*$), F($y^*$), F($z^*$)),where (F($x^*$), F($y^*$), F($z^*$)) satisfies the following simultaneous equations:

$$\begin{cases} C_H - C_L - zR - yV + bzR + ayV = 0 \\ C_g - R_g - F - V + aV + xF + xV - axV = 0 \\ C_s - R + bR + xR - bxR = 0 \end{cases} \tag{14}$$

## 3. Evolutionary stability analysis

The equilibrium points of the evolutionary game can be derived from the Jacobian matrix. In this paper, the Jacobian matrix corresponding to the evolutionary game model of the PPP project is as follows:

$$\text{Jocobian matrix} = \begin{Bmatrix} \frac{\partial F(x)}{\partial(x)} & \frac{\partial F(x)}{\partial(y)} & \frac{\partial F(x)}{\partial(z)} \\ \frac{\partial F(y)}{\partial(x)} & \frac{\partial F(y)}{\partial(y)} & \frac{\partial F(y)}{\partial(z)} \\ \frac{\partial F(z)}{\partial(x)} & \frac{\partial F(z)}{\partial(y)} & \frac{\partial F(z)}{\partial(z)} \end{Bmatrix} = \tag{15}$$

$$\begin{Bmatrix} (2x - 1)(C_H - C_L - (1 - bz)R - (1 - a)yV) & x(1 - x)(1 - a)V & x(1 - x)(1 - b)R \\ y(y - 1)(F + (1 - a)V) & (2y - 1)\big(C_g - R_g - (1 - x)F - (1 - x)(1 - a)V\big) & 0 \\ z(z - 1)(1 - b)R & 0 & (2z - 1)(C_s - (1 - x)(1 - b)R) \end{Bmatrix}$$

In asymmetric evolutionary games, if the equilibrium point of the evolutionary game is evolutionarily stable, then the equilibrium must be a strict Nash equilibrium, which is a pure strategy equilibrium. In asymmetric evolutionary games, the mixed strategy equilibrium must not be evolutionarily stable, so the asymptotic stability of the pure strategy equilibrium should be discussed only, such as (0,0,0), (0,0,1), (0,1,0), (0,1,1), (1,0,0), (1,1,0), (1,0,1), (1,1,1). According to Lyapunov Stability Theory [33–35],the equilibrium point is an ESS(Evolutionary Stable Strategy) only if all three eigenvalues of the Jacobian matrix are negative.

When all eigenvalues of the Jacobian matrix are positive, the equilibrium point is an unstable point. When there are positive and negative eigenvalues of the Jacobian matrix, the equilibrium point is an unstable point, which is a saddle point. By substituting the eight equilibrium points into (15), the eigenvalues of the Jacobian matrix can be calculated as shown in Table 3.

**Table 3. Eigenvalues of the Jacobian matrix.**

| Equilibrium points | Eigenvalue 1 | Eigenvalue 2 | Eigenvalue 3 | Stability | Stable condition |
|---|---|---|---|---|---|
| (0,0,0) | $C_L - C_H$ | $R - C_s - b * R$ | $R_g - C_g + F + (1-a) * V$ | ESS | $(1-b) * R < C_s,$ $R_g + F + (1-a) * V < C_g$ |
| (1,0,0) | $C_H - C_L$ | $R_g - C_g$ | $-C_s$ | unstable | |
| (0,1,0) | $R - C_s - b * R$ | $C_L - C_H + (1-a) * V$ | $C_g - R_g - F - (1-a) * V$ | ESS | $(1-b) * R < C_s,$ $(1-a) * V < C_H - C_L,$ $R_g - C_g + F + (1-a) * V > 0$ |
| (0,0,1) | $C_s - (1-b) * R$ | $C_L - C_H + (1-b) * R$ | $R_g - C_g + F - (1-a) * V$ | ESS | $(1-b) * R > C_s,$ $(1-b) * R < C_H - C_L,$ $R_g - C_g + F + (1-a) * V < 0$ |
| (1,1,0) | $C_g - R_g$ | $-C_s$ | $C_H - C_L - (1-a) * V$ | ESS | $R_g > C_g,$ $(1-a) * V > C_H - C_L$ |
| (1,0,1) | $C_s$ | $R_g - C_g$ | $C_H - C_L - (1-b) * R$ | ESS | $C_s < 0,$ $R_g < C_g,$ $(1-b) * R > C_H - C_L$ |
| (0,1,1) | $C_s - (1-b) * R$ | $C_g - R_g - F - (1-a) * V$ | $C_L - C_H + (1-b) * R + (1-a) * V$ | ESS | $(1-b) * R > C_s,$ $R_g - C_g + F + (1-a) * V < 0,$ $C_H - C_L > (1-b) * R + (1-a) * V$ |
| (1,1,1) | $C_s$ | $C_g - R_g$ | $C_H - C_L - (1-b) * R - (1-a) * V$ | ESS | $C_s < 0,$ $R_g > C_g,$ $C_H - C_L < (1-b) * R + (1-a) * V$ |

1. When $C_s > (1-b)R$, $C_g > R_g + F + (1-a)V$, the eigenvalues of the Jacobian matrix corresponding to the evolutionary stable point $(0, 0, 0)$ are all negative, indicating that point $(0, 0, 0)$ is an ESS. This suggests that i) The cost of consumer supervision is greater than the price deduction that consumers derive from monitoring poor quality services, and ii) The cost of government regulation is greater than the total of benefits, reputation, and deductions from government subsidies to the project. The strategy profile (low-quality, non-regulation, non-supervision) is an ESS.

2. When $C_s > (1-b)R$, $(1-a)V < C_H - C_L$ and $R_g + F + (1-a)V > C_g$, the eigenvalues of the Jacobian matrix corresponding to the evolutionary stable point $(0, 1, 0)$ are all negative, indicating point $(0, 1, 0)$ is an ESS. This suggests that i)the cost of consumer supervision is greater than the price deduction that consumers derive from monitoring poor-quality services, ii)the difference in cost between high-quality services and low-quality services is greater than the amount of government subsidies deducted from social capital, iii)the cost of government regulation is smaller than the total of benefits, reputation, and deductions from government subsidies. The strategy profile (low-quality, regulation, non-supervision) is an ESS.

3. When $(1-b)R > C_s$, $(1-b)R < C_H - C_L$ and $R_g + F + (1-a)V < C_g$, the eigenvalues of the Jacobian matrix corresponding to the evolutionary stable point $(0, 0, 1)$ are all negative, indicating that point $(0, 0, 1)$ is an ESS. This suggests that i)the cost of consumer supervision is smaller than the price reduction that consumers obtain from social capital by using low-quality services, ii) The difference in cost between high-quality services and low-quality services is greater than the price deduction that consumers obtain from social capital by using low-quality services, iii) The cost of government regulation is greater than the total revenue, reputation, and deduction of government subsidies by the government in regulation. The strategy profile (low-quality, non-regulation, supervision) is an ESS.

4. When $R_g > C_g$, $(1-a)V > C_H - C_L$, the eigenvalues of the Jacobian matrix corresponding to the evolutionary stable point $(1, 1, 0)$ are all negative, indicating point $(1, 1, 0)$ is an ESS.

This suggests that i)the benefits of government regulation outweigh the costs, and ii)the difference of cost between high-quality services and low-quality services is smaller than the amount of government subsidies deducted from social capital. The strategy profile (high-quality, regulation, non-supervision) is an ESS.

5. When $C_s < 0$, $R_g < C_g$, $(1 − b)R > C_H − C_L$, the eigenvalues of the Jacobian matrix corresponding to the evolutionary stable point $(1, 0, 1)$ are all negative, indicating point $(1, 0, 1)$ is an ESS. This suggests that i)the costs of consumer supervision are negative. ii) the benefit of government regulation is less than the regulatory cost; iii) the difference in cost between high-quality services and low-quality services is less than the price reduction brought by consumers from the supervision of low quality services. The strategy profile (high-quality, non-regulation, supervision) is an ESS.

6. When $(1 − b)R > C_s$, $R_g + F + (1 − a)V > C_g$, $C_H − C_L > (1 − b)R + (1 − a)V$, the eigenvalues of the Jacobian matrix corresponding to the evolutionary stable point $(0, 1, 1)$ are all negative, indicating point $(0, 1, 1)$ is an ESS. This suggests that i)The cost of consumers supervision is smaller than the price reduction that consumers obtain from social capital by the low-quality services, ii)The cost of the government regulation is smaller than the total of revenue, reputation and deduction of government subsidies of the government in regulation. iii)The difference of cost between high quality services and low quality services is greater than the sum of the price reduction brought by consumers in supervising low-quality services and the subsidy reduction brought by the government regulation in low-quality services. The strategy profile (low-quality, regulation, supervision) is an ESS.

7. When $C_s < 0$, $R_g < C_g$, $C_H − C_L < (1 − b)R + (1 − a)V$, the eigenvalues of the Jacobian matrix corresponding to the evolutionary stable point $(1, 1, 1)$ are all negative, indicating point $(1, 1, 1)$ is an ESS. This suggests that i)the costs of consumers supervision are negative. ii)the benefit of government in regulation is less than the regulatory cost, iii)the difference of cost between high-quality services and low quality services is greater than the sum of the price reduction brought by consumers in supervising low-quality services and the subsidy reduction brought by government regulation in low-quality services. The strategy profile (high-quality, regulation, supervision) is an ESS.

## 4. Numerical simulation

### 4.1. Evolutionary path of ESS

Based on the replication dynamic equation of the tripartite evolutionary game, this section employ MATLAB to simulate the evolutionary path of the seven ESS points mentioned in section 3.

1) Assume, $R = 60$, $R_g = 10$, $C_g = 20$, $C_H = 50$, $C_L = 40$, $V = 20$, $a = 0.9$, $b = 0.9$, $F = 5$, $C_s = 8$, the evolutionary path of $(0, 0, 0)$ is demonstrated in Fig 1(a). When the initial willingness $x = 0.5$, $y = 0.5$, and $z = 0.5$, the evolutionary path of $(0, 0, 0)$ is illustrated in Fig 1(b).

2) Assume, $R = 60$, $R_g = 10$, $C_g = 15$, $C_H = 50$, $C_L = 40$, $V = 20$, $a = 0.9$, $b = 0.9$, $F = 5$, $C_s = 8$, the evolutionary path of $(0, 1, 0)$ is demonstrated in Fig 2(a). When the initial willingness $x = 0.5$, $y = 0.5$, and $z = 0.5$, the evolutionary path of $(0, 1, 0)$ is illustrated in Fig 2(b).

3) Assume, $R = 60$, $R_g = 10$, $C_g = 20$, $C_H = 50$, $C_L = 40$, $V = 20$, $a = 0.9$, $b = 0.9$, $F = 5$, $C_s = 4$, the evolutionary path of $(0, 0, 1)$ is demonstrated in Fig 3(a). When the initial willingness $x = 0.5$, $y = 0.5$, and $z = 0.5$, the evolutionary path of $(0, 0, 1)$ is illustrated in Fig 3(b).

(a)

(b)

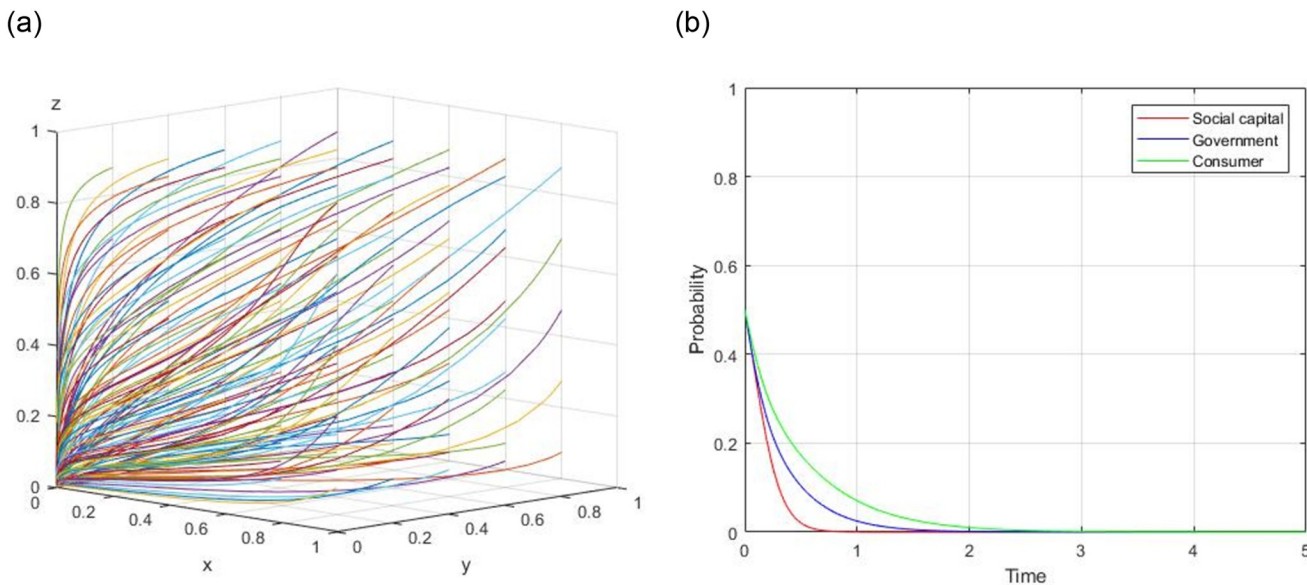

**Fig 1. Evolutionary path of (0, 0, 0).**

4) Assume, $R = 60$, $R_g = 20$, $C_g = 10$, $C_H = 50$, $C_L = 40$, $V = 20$, $a = 0.4$, $b = 0.9$, $F = 5$, $C_s = 4$, the evolutionary path of $(1, 1, 0)$ is demonstrated in Fig 4(a). When the initial willingness $x = 0.5$, $y = 0.5$, and $z = 0.5$, the evolutionary path of $(1, 1, 0)$ is illustrated in Fig 4(b).

5) Assume, $R = 60$, $R_g = 10$, $C_g = 20$, $C_H = 50$, $C_L = 40$, $V = 20$, $a = 0.4$, $b = 0.8$, $F = 5$, $C_s = -3$, the evolutionary path of $(1, 0, 1)$ is demonstrated in Fig 5(a). When the initial willingness $x = 0.5$, $y = 0.5$, and $z = 0.5$, the evolutionary path of $(1, 0, 1)$ is illustrated in Fig 5(b).

6) Assume, $R = 60$, $R_g = 20$, $C_g = 10$, $C_H = 50$, $C_L = 40$, $V = 20$, $a = 0.8$, $b = 0.8$, $F = 5$, $C_s = 4$, the evolutionary path of $(0, 1, 1)$ is demonstrated in Fig 6(a). When the initial willingness $x = 0.5$, $y = 0.5$, and $z = 0.5$, the evolutionary path of $(0, 1, 1)$ is illustrated in Fig 6(b).

(a)

(b)

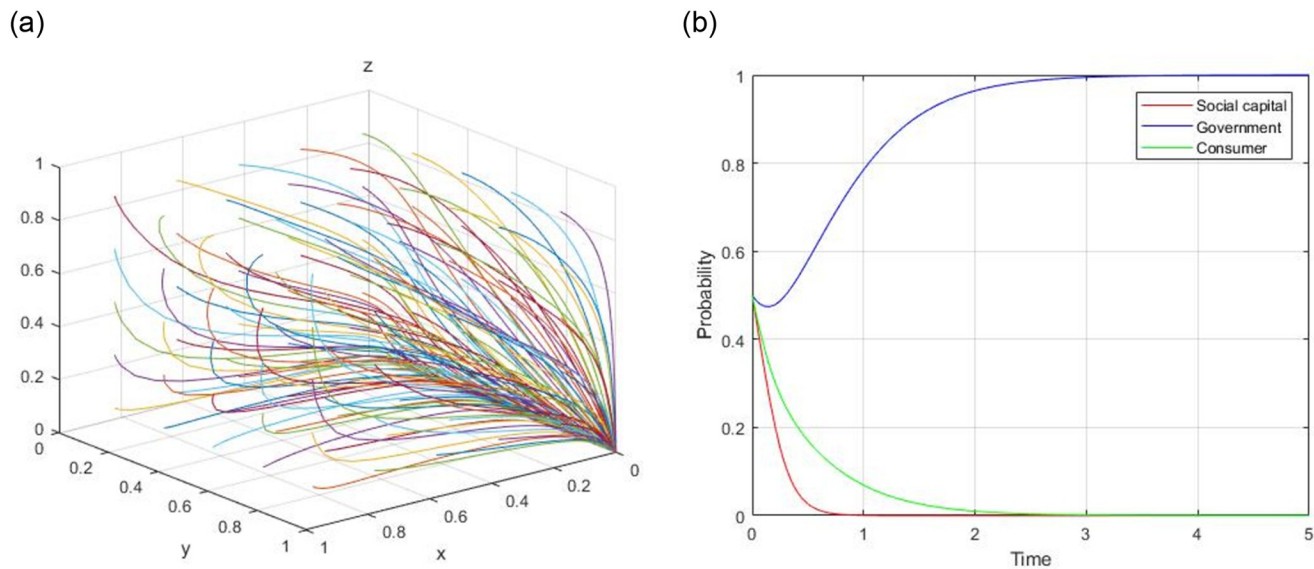

**Fig 2. Evolutionary path of (0, 1, 0).**

(a) 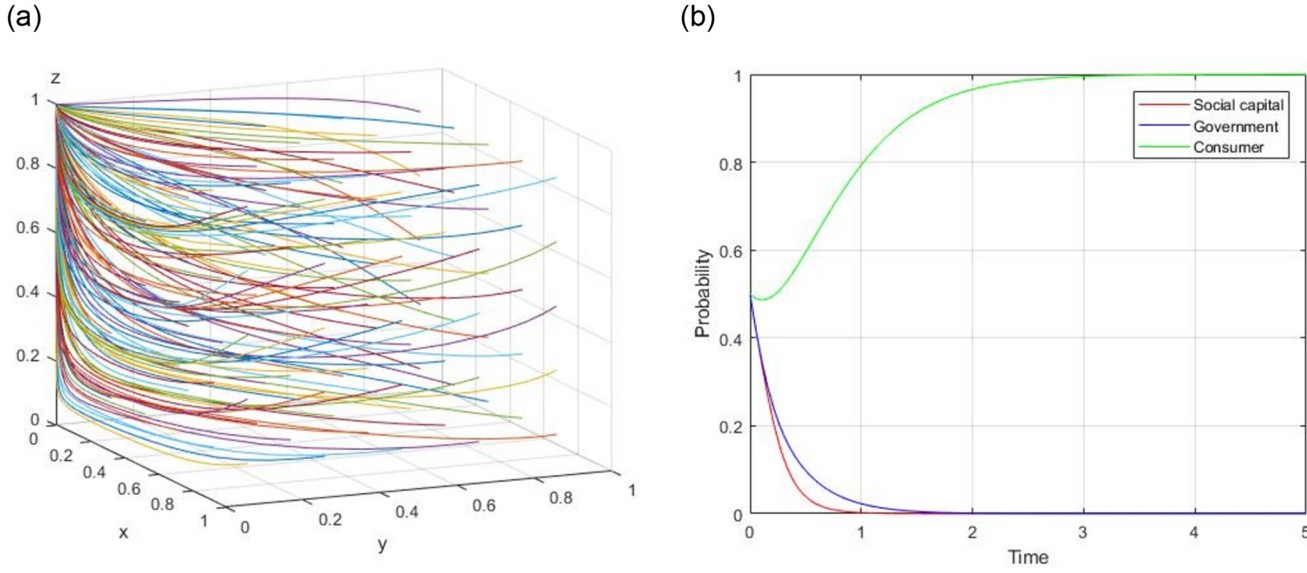 (b)

**Fig 3. Evolutionary path of (0, 0, 1).**

7) Assume, $R = 60$, $R_g = 20$, $C_g = 10$, $C_H = 50$, $C_L = 40$, $V = 20$, $a = 0.4$, $b = 0.8$, $F = 5$, $C_s = -3$, the evolutionary path of (1, 1, 1) is demonstrated in Fig 7(a). When the initial willingness $x = 0.5$, $y = 0.5$, and $z = 0.5$, the evolutionary path of (1, 1, 1) is illustrated in Fig 7(b).

## 4.2. Impacts of key parameters on evolutionary results and trajectories

In order to further analyze the impact of some key parameters in the tripartite evolutionary game, a numerical simulation based on a scenario is conducted in this section. The initial values of all parameters based on the reality scenario are: $R = 60$, $R_g = 20$, $C_g = 10$, $C_H = 50$, $C_L =$

(a) 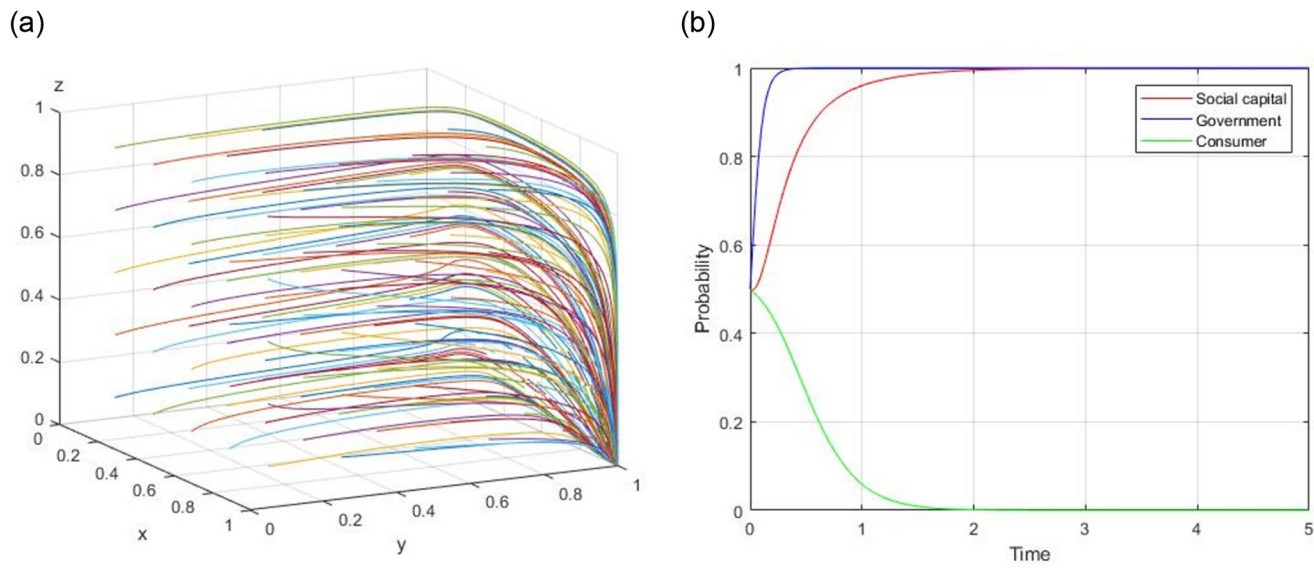 (b)

**Fig 4. Evolutionary path of (1, 1, 0).**

(a)

(b)

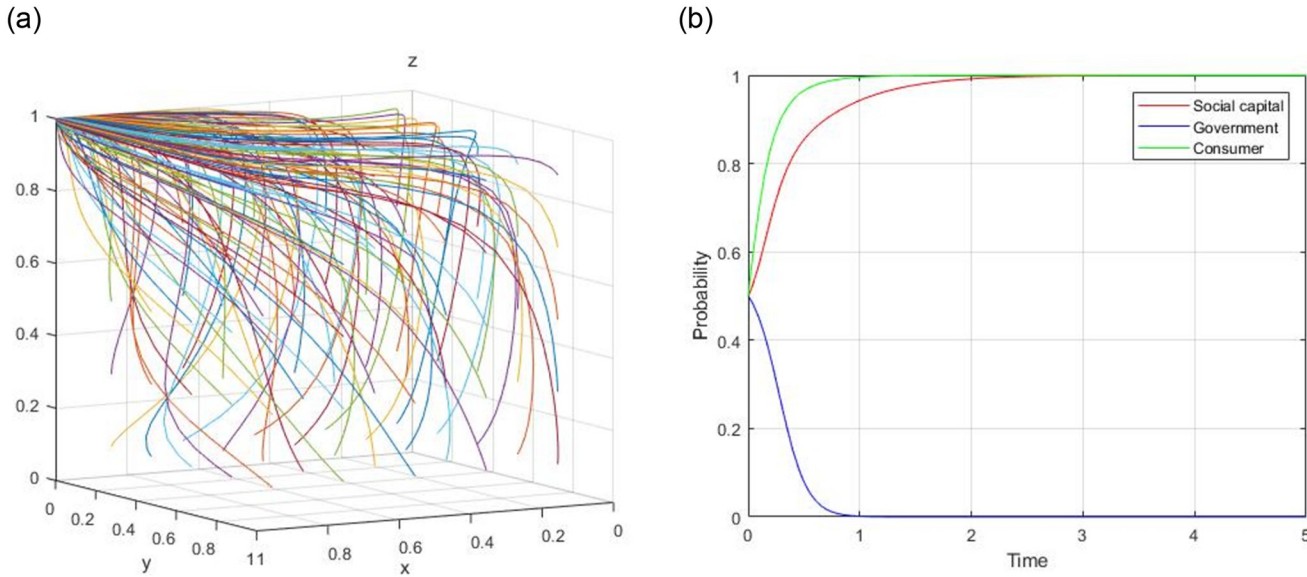

**Fig 5. Evolutionary path of (1, 0, 1).**

40, V = 20, a = 0.9, b = 0.9, F = 5, $b_1$ = 20, $b_2$ = 10, $C_s$ = 4. According to the analysis in Table 3, the dynamic processes are sensitive to some parameters, including $C_H - C_L$, a, b, $R_g - C_g$, $C_s$. Meanwhile, we assume a scenario where the initial probability of all participants strategy adoption is set as 0.5. Thus, the impacts of the key parameters on evolutionary results and the path of social capital's strategy, the government's strategy, and the consumer's strategy are discussed as follows.

**4.2.1 impact analysis of the difference between $C_H$ and $C_L$.** Let $C_H - C_L$ = 0, 5, and 10 respectively, while other parameters are unchanged. The impact of $C_H - C_L$ on evolutionary

(a)

(b)

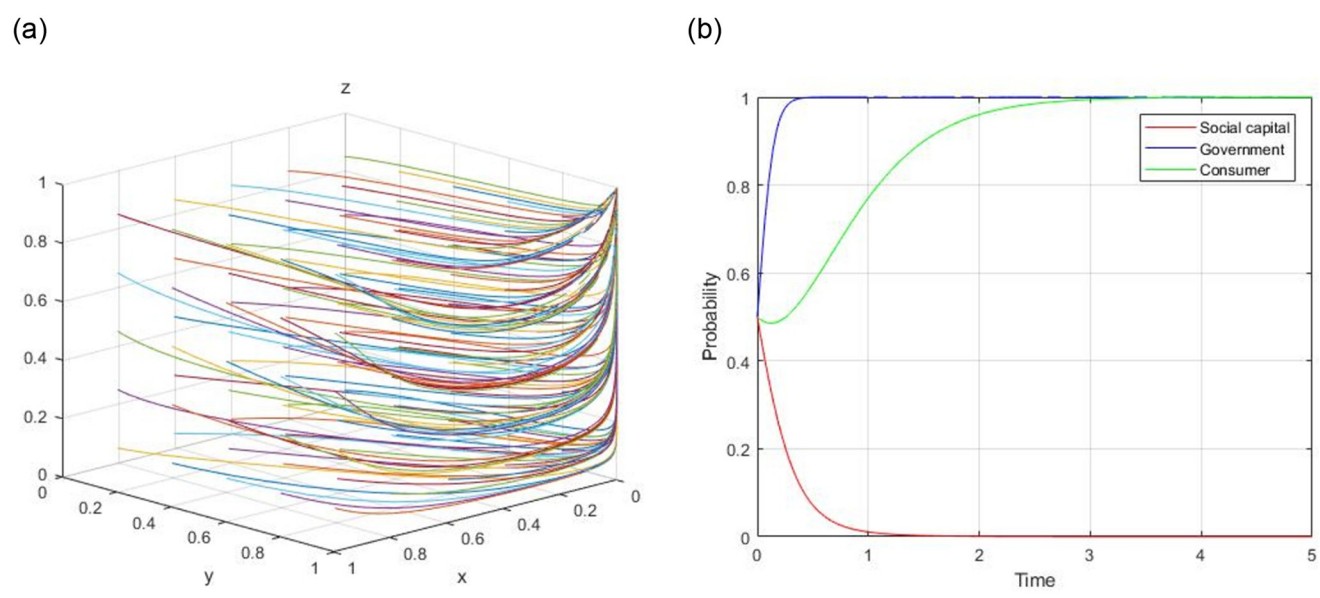

**Fig 6. Evolutionary path of (0, 1, 1).**

(a)                                                                    (b)

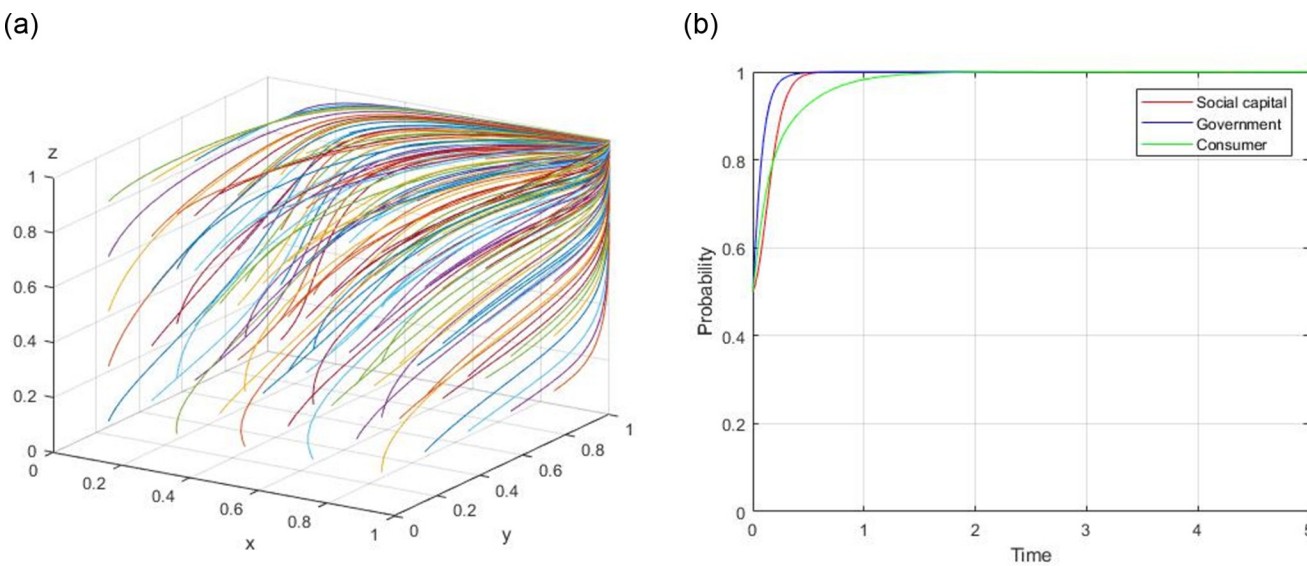

**Fig 7. Evolutionary path of (1, 1, 1).**

results and paths is shown in Fig 8. The difference between $C_H$ and $C_L$ represents the different costs of different strategies of social capital. When the difference between $C_H$ and $C_L$ is 10, the evolutionary stable point is (0,1,1). As the difference between $C_H$ and $C_L$ decreases, the difference in service quality is relatively small. Social capital tends to provide high-quality services, and consumers tend to adopt a non-supervision strategy. The evolutionary stable point is (1,1,0). So raising the standard of low-quality services can prompt social capital to provide high-quality services. It is not necessary for consumers to be involved in supervision. The government's strategy won't be influenced by the change in service costs difference.

**4.2.2 impact analysis of 'a'.** Let $a$ = 0.3, 0.7, and 0.9 respectively, while other parameters are unchanged. The impact of the subsidy coefficient for government on social capital on evolutionary results and paths is shown in Fig 9. The parameter 'a' represents the subsidy coefficient for government to social capital. When the initial value of 'a' is from 0.9 to 0.3, the evolutionary stable point is from (0,1,1) to (1,1,0). This paper assumes that V represents the government's full financial subsidy to social capital. Then a*V represents the government's real financial subsidy to social capital according to the service quality provided by social capital. When the parameter 'a' is 0.3, social capital will choose to supply high-quality services in order to avoid subsidy reduction. When the parameter 'a' is 0.9, social capital will choose to supply low-quality services. So the parameter 'a' imposes a significantly negative impact on social capital's strategy. The parameter 'a' has no impact on the government's regulation strategy. For consumers, the parameter 'a' imposes a positive impact on their strategy. When 'a' is 0.9, consumers choose the supervision strategy due to social capital's low-quality strategy. When 'a' is 0.3, consumers choose a non-supervision strategy due to their speculative attitude.

**4.2.3 impact analysis of 'b' and $C_s$.** According to the model assumption, there are two parameters that affect the consumer's decision, which are 'b' and $C_s$. The impact of changes in 'b' on consumer decisions should be analyzed in the case of $C_s > 0$ and $C_s < 0$.

1) When $C_s > 0$, the cost of consumer' supervision is positive. Let $C_s$ = 4, $b$ = 1, 0.9, and 0.8 respectively. While other parameters are unchanged, the impact of the price coefficient for consumer to social capital on evolutionary results and paths is shown in Fig 10. When the

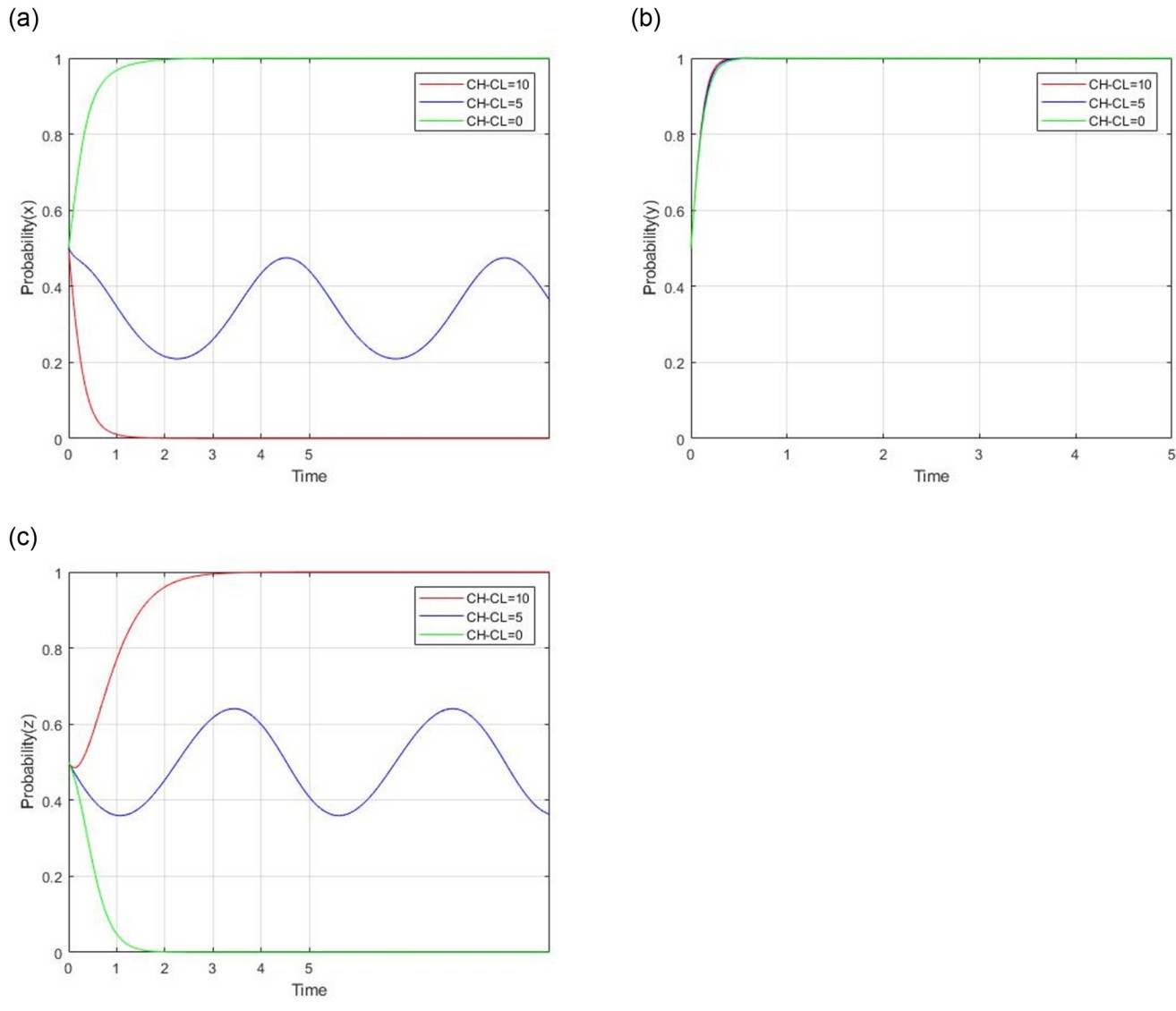

**Fig 8. Impact of the difference between $C_H$ and $C_L$.**

initial values of 'b' are 1 and 0.9, the evolutionary stable points are (0,1,0) and (0,1,1),respectively. A lower 'b' can encourage consumers to choose a supervision strategy. Social capital always chooses a low-quality service strategy. The government chooses a regulation strategy. But while $b = 0.8$, the revolutionary game can not evolve into a stable point.

2) When $C_s < 0$, the cost of consumer supervision is negative. Let $C_s = -1$, $b = 1$, 0.9 and 0.8 respectively, while other parameters are unchanged, the impact of the price coefficient for consumer to social capital on evolutionary results and paths is shown in Fig 11. Generally speaking, the evolutionary stable point (0,1,1) will evolveinto (1,1,1) as 'b' decreases. For social capital, changes in 'b' exert a significantly negative impact on social capital's high-quality strategy. When b = 1 and 0.9, social capital will choose the low-quality strategy due to the smaller revenue impact. When b = 0.8,social capital will prefer to supply high-quality service due to the lower discount offer to consumers by social capital. The government chooses regulation

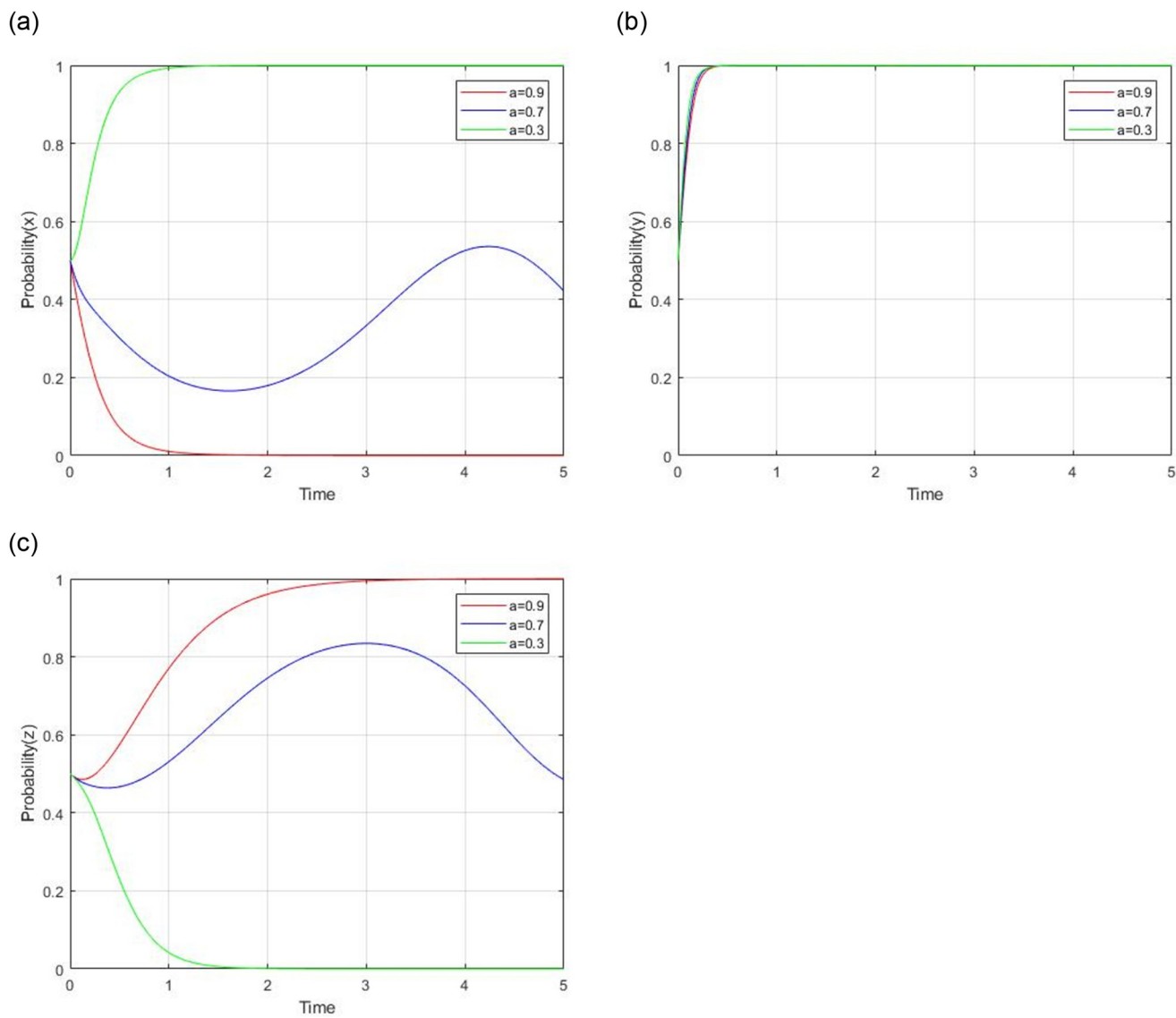

**Fig 9. Impact of subsidy coefficient for government to social capital.**

strategy. For consumer, a lower discount will lead to its stronger and faster willingness of consumer's supervision strategy.

Generally, $C_s$ can influence consumer's strategic decision. When $C_s > 0$, consumers tend to adopt non-supervision. Under certain conditions on other parameters, consumers will a choose supervision strategy. When $C_s < 0$, consumers tend to adopt a supervision strategy.

**4.2.4 impact analysis of $R_g - C_g$.** Regulation benefits of government is set as $R_g$ and regulation costs of government is set as $C_g$. According to the equilibrium condition in Table 3, $R_g - C_g$ will be analyzed as a whole. Let $R_g = 20$ and $C_g = 10$, $R_g = 10$ and $C_g = 10$, $R_g = 0$ and $C_g = 10$, respectively, while other parameters are unchanged, the impact of the difference between the benefit and cost of government on evolutionary results and paths is shown in Fig 12. When $R_g - C_g = 10$, the government's regulatory benefits outweigh the regulatory costs. The government tends to be involved in regulation. When $R_g - C_g = -10$, the government's regulatory

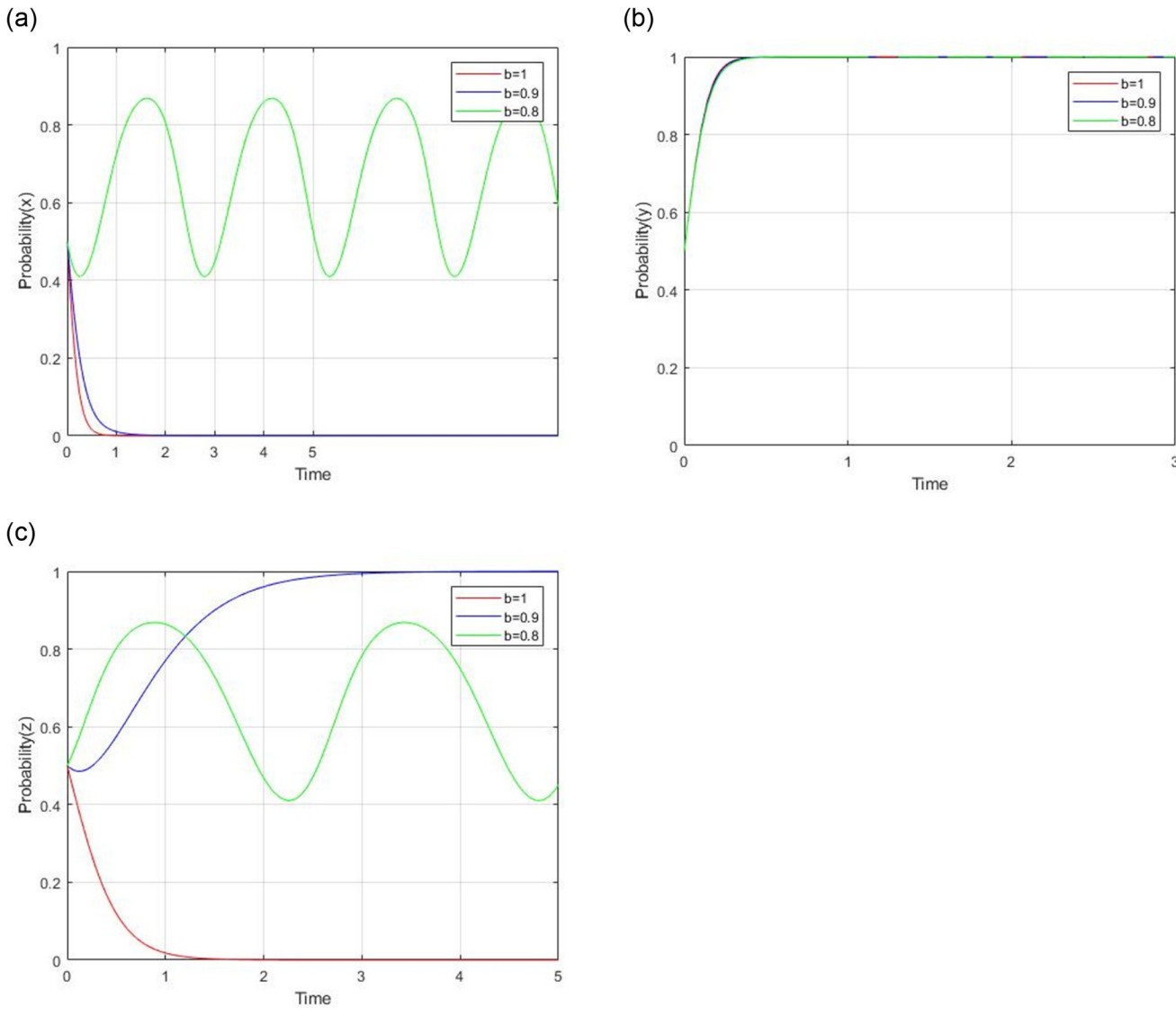

**Fig 10. Impact of subsidy coefficient for government to social capital when $C_s > 0$.**

benefits are less than the regulatory costs. The government tends to adopt a non-regulation strategy. While $R_g - C_g$ is from 10 to -10, the ESS of the revolutionary game is from (0,1,1) to (0,0,1). The difference between the benefit and cost of government won't change the strategy of social capital and consumers. It will only change the strategy of the government from regulation to non-regulation.

**4.2.5 impact analysis of initial willingness.** Let $x_0$ = 0.9,0.5,0.2, $y_0$ = 0.9,0.5,0.2, $z_0$ = 0.9,0.5,0.2, while keeping other parameters unchanged, the impact of initial willingness on evolutionary results and path is illustrated in Fig 13. The evolutionary stable point (0, 1, 1) remains unchanged, whatever the initial willingness is. It indicates that the initial willingness doesn't affect the evolutionary results, but different initial values will affect the path to equilibrium. For social capital, a negative impact of $x_0$ on social capital's low-quality service strategy is shown. A lower initial value will increase their probability of choosing the low-quality service

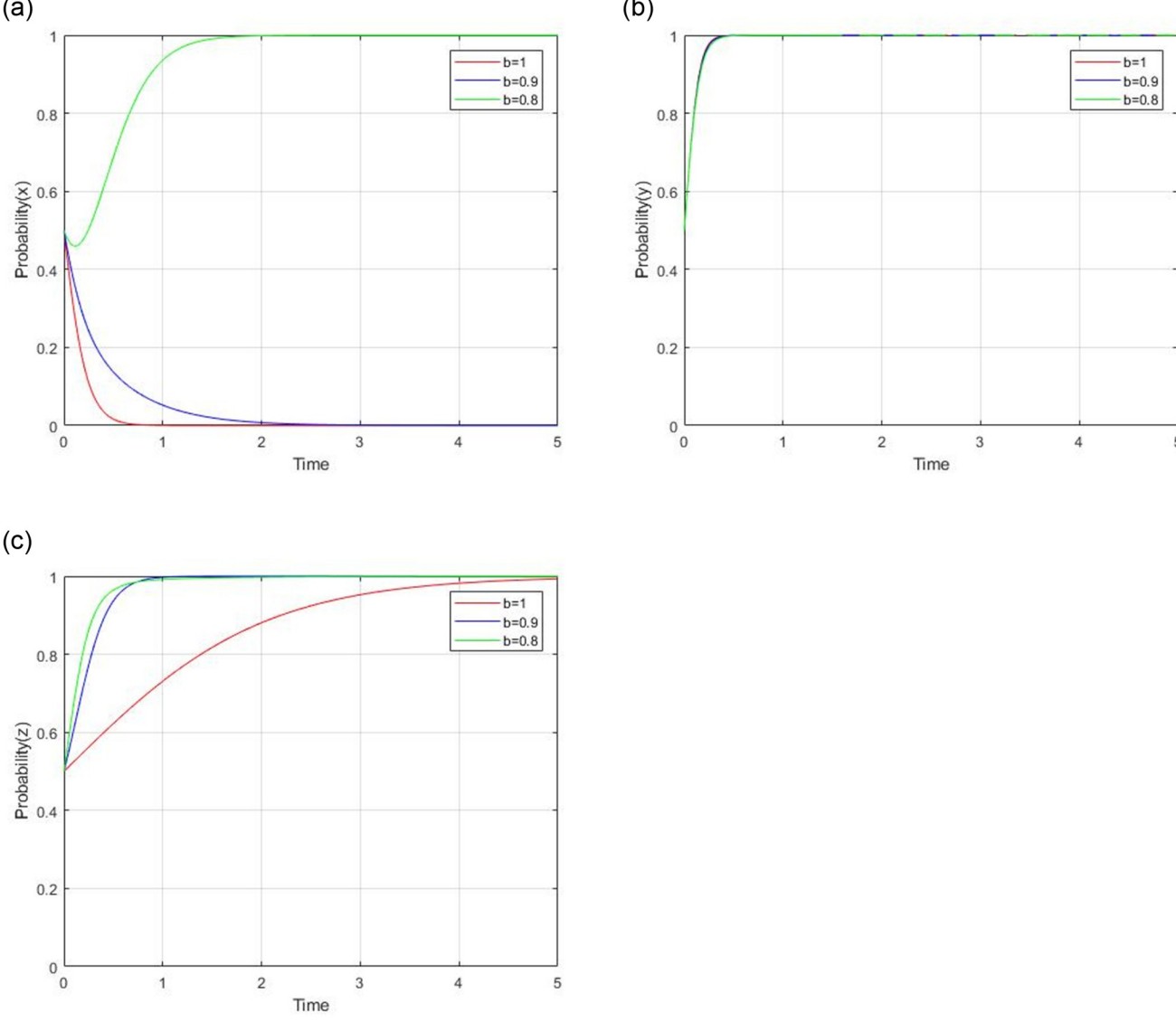

**Fig 11. Impact of subsidy coefficient for government to social capital when $C_s < 0$.**

strategy and shorten the time to reach the evolutionary stable point. For government, a positive impact of $y_0$ on the government's regulation strategy is shown. A higher initial value will increase their probability of choosing the regulation strategy. For consumers, the impact of different initial values of $z_0$ is not monotonous. But consumers will adopt the supervision strategy, whatever the initial value.

## 5. Conclusions and policy implications

### 5.1. Conclusion

In this work, the tripartite evolutionary game model is utilized to investigate the ESSs of social capital, government, and consumer in order to show the behavior of consumer participation in the PPP model. By looking at the asymptotic stability of three parties separately, seven ESSs

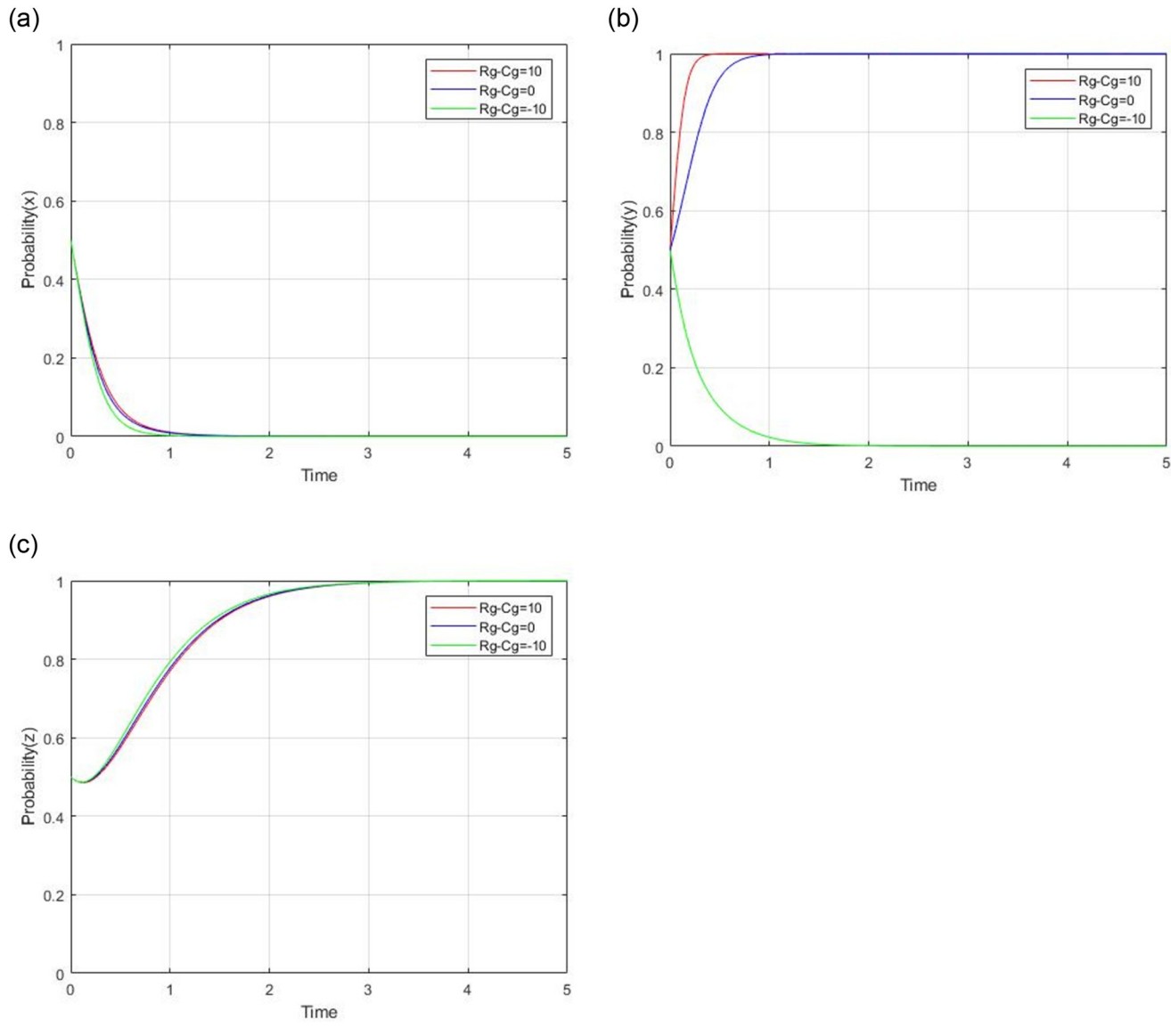

**Fig 12. Impact of the difference between $R_g$ and $C_g$.**

and their associated stable conditions are identified. To demonstrate the seven ESSs as well as the effects of several important parameters on the tripartite revolutionary game, a simulation is employed. The following is a summary of the key findings.

First, the stable circumstances of a tripartite decision are determined based on the examination of seven ESSs and their corresponding stable conditions. The consumer's choice of strategy is determined by a comparison of the consumer's cost of supervision and the price coefficient of their social capital. The choice of social capital is decided by comparing the subsidy coefficient for the government to social capital and the difference between high-quality cost and low-quality cost when consumers pick the non-supervision strategy and the government adopts the regulatory strategy. No matter what approach the government adopts, when consumers adopt the monitoring strategy, the strategy of social capital is decided by the

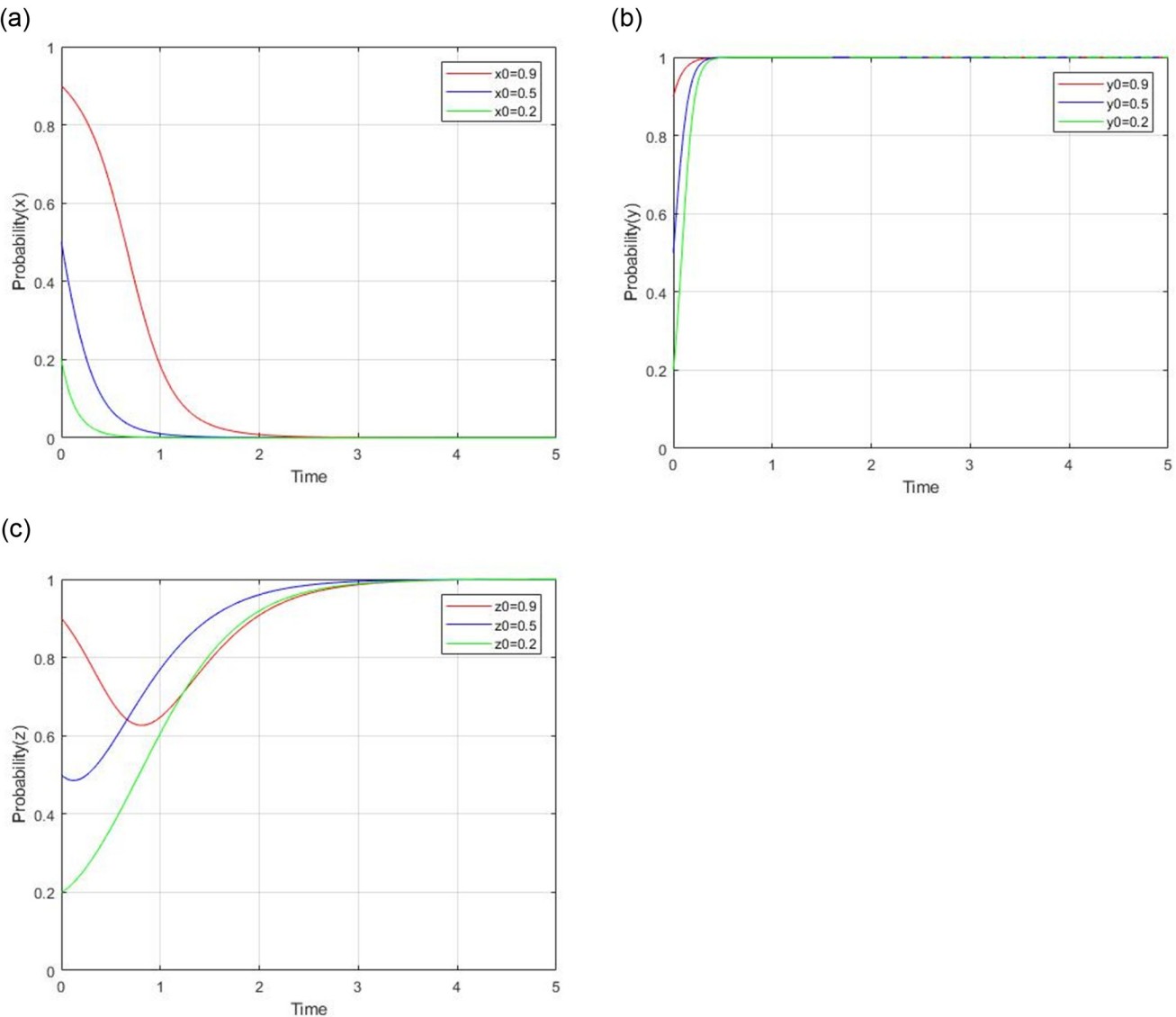

**Fig 13. Impact of initial willingness on evolutionary results and path.**

comparison of the gap between high-quality cost and low-quality cost as well as the price coefficient for consumers to social capital. One prerequisite is that the consumer's supervision cost is negative before social capital chooses the high-quality approach and consumers adopt the supervision strategy.

Second, the impacts of key parameters are noted on the tripartite evolutionary game. The difference between high-quality and low-quality costs as well as the government's subsidy coefficient directly influence the choice of social capital. The decision of the customer is directly influenced by the supervision cost of the consumer and the price coefficient of the consumer's social capital. Government decisions are exclusively impacted by the difference between benefit and cost; neither consumers nor social capital are impacted. The tripartite evolutionary game finds equilibrium more quickly or more slowly depending on the initial desire.

### 5.2. Policy implication

In this paper, we establish the tripartite evolutionary game model to simulate the strategies of social capital, government, and consumers. We evaluate the seven ESSs in the simulation and examine the influence of five parameters on the choice of tripartite strategies. It is discovered that five parameters affect social capital, the government, and consumers in different ways. Based on the aforementioned findings, policy recommendations are made to encourage consumer participation and social capital building to deliver high-quality services under less restrictive regulatory conditions.

From the consumer's perspective, It is initially important to lower the cost of consumer participation in supervision and increase the effectiveness of consumer supervision on service quality. The three parties should work together to develop a realistic and normative standard of quality evaluation in order to increase the effectiveness of consumer oversight. Second, a price reduction provision for variations in service quality should be included in the service agreement that consumers and social capital sign. Price cuts should be more substantial than the cost differences between high- and low-quality services. The deduction amount should also be higher than the expense of consumer supervision at the same time. Finally, through social capital, consumer supervision can encourage the delivery of high-quality services. In order to encourage customers to take part in oversight, the government might provide them with the proper subsidies. The quantity of subsidies must be higher than the price of providing consumer supervision.

From the government's perspective, a benchmark should be established for assessing the caliber of the services rendered by social capital. The various service quality levels can be precisely identified by the standard. In addition, the quantity of government financial subsidies and service quality will be related. Subsidies for high- and low-quality services ought to differ significantly. Fiscal subsidies might be set up differently to encourage social capital and high-quality services. Second, the cost of the regulation must be borne by the government. Governments may participate more actively in regulating if there are lower costs associated with regulation. Government active regulation can also encourage social capital to deliver high-quality services in order to satisfy customer needs. As a result, the government should actively increase management efficiency and lower regulatory expenses.

In conclusion, strong government regulation alone can encourage social capital to offer high-quality services, but the government must get involved in more areas. Promoting social capital, on the other hand, is not achievable if high-quality services are only provided under customers supervision. Government involvement is crucial for the creation of standards for the quality of services, consumer engagement subsidies, and other initiatives. From the standpoint of public governance, the government should diversify its ways of governing and promote widespread public participation. The active participation of consumers can support the sustainable growth of PPP ventures when the government provides uniform instructions.

## Supporting information

**S1 Data. Minimal data.**
(DOCX)

## Author Contributions

**Conceptualization:** Wei Liu.

**Data curation:** Wei Liu.

**Formal analysis:** Wei Liu.

**Investigation:** Wei Liu.

**Methodology:** Sheng Jiang.

**Project administration:** Xiaoli Wang.

**Resources:** Wei Liu.

**Software:** Wei Liu.

**Visualization:** Wei Liu.

**Writing – original draft:** Wei Liu.

**Writing – review & editing:** Wei Liu.

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
