## [Decision Letter · Decision Letter 0]

12 Dec 2023

PONE-D-23-19775Decision Analysis of PPP Project’s Parties Based on Deep Consumer ParticipationPLOS ONE

Dear Dr. Liu,

Thank you for submitting your manuscript to PLOS ONE. After careful consideration, we feel that it has merit but does not fully meet PLOS ONE’s publication criteria as it currently stands. Therefore, we invite you to submit a revised version of the manuscript that addresses the points raised during the review process.

I strongly encourage the authors to address all the issues kindly raised by the reviewers and in particular, I strongly advise a thourough proofreading of the paper which at the moment is extremely difficult to read. moreover, its positioning and contribution must be stated more clearly, the methodology explained better and the discussion of the results must be conducted more in depth clearly pointing out the addedd value of the analysis.

We look forward to receiving your revised manuscript.

Kind regards,

Maurizio Fiaschetti

Academic Editor

PLOS ONE

Journal Requirements:

Reviewers' comments:

Reviewer's Responses to Questions

**Comments to the Author**

1. Is the manuscript technically sound, and do the data support the conclusions?

Reviewer #1: Yes

Reviewer #2: Yes

2. Has the statistical analysis been performed appropriately and rigorously? 

Reviewer #1: No

Reviewer #2: Yes

3. Have the authors made all data underlying the findings in their manuscript fully available?

Reviewer #1: Yes

Reviewer #2: Yes

4. Is the manuscript presented in an intelligible fashion and written in standard English?

Reviewer #1: No

Reviewer #2: Yes

5. Review Comments to the Author

Reviewer #1: Please see attached. Most importantly, writing needs to be improved. Second, the reference list needs to be better organized.

Please see attached. Most importantly, writing needs to be improved. Second, the reference list needs to be better organized.

Reviewer #2: This paper is interesting and find some valuable conclusion. The paper can be accepted after making the following revisions.

1. The innovation of this paper needs to be highlighted in the abstract.

2. The author introduced the research background of this paper too much, but they does not explain

3. The literature review is not enough, the innovation of this paper and the contribution made by previous studies have not been clearly expressed. The innovation of this paper and the contribution made by previous studies have not been clearly expressed. The author should summarize the existing research gaps and highlight the innovation of this paper after completing the literature review.

4. The presentation of the method is very imprecise

5. Interpretation of results does not highlight important issues studied in this paper

6. The discussion section does not analyze the results of this paper, and provides enlightening thinking on related issues. the following literature should be helpful for your research：(1) Reduction pathways identification of Agricultural Water Pollution in Hubei Province, China. (2) Coordination of the Industrial-Ecological Economy in the Yangtze River Economic Belt, China.

7. Compared with the available literature, what are the theoretical contributions and application values of this study? It is suggested to enhance the corresponding discussions in the conclusion part.

6. PLOS authors have the option to publish the peer review history of their article (what does this mean?). If published, this will include your full peer review and any attached files.

Reviewer #1: No

Reviewer #2: No

---

## [Author Response · Author response to Decision Letter 0]

26 Jan 2024

Reviewer #1: Please see attached. Most importantly, writing needs to be improved. Second, the reference list needs to be better organized.

Response:I have checked the grammar and words in the whole text and made changes to make the sentences clearer, concise and readable.

Please see attached. Most importantly, writing needs to be improved. Second, the reference list needs to be better organized.

Response:I have checked the reference and reorganized it.

Reviewer #2: This paper is interesting and find some valuable conclusion. The paper can be accepted after making the following revisions.

1. The innovation of this paper needs to be highlighted in the abstract.

Response :I have made revisions to the abstract, highlighting the innovation of the article and emphasizing the significant impact of consumer's active involvement on decision-making processes within the project.

2. The author introduced the research background of this paper too much, but they does not explain

Response :The language in the background section has been reorganized, and unnecessary content has been removed.

3. The literature review is not enough, the innovation of this paper and the contribution made by previous studies have not been clearly expressed. The innovation of this paper and the contribution made by previous studies have not been clearly expressed. The author should summarize the existing research gaps and highlight the innovation of this paper after completing the literature review.

Response :I have reorganized the existing literature review by providing an overview of the content in the first sentence of each paragraph. Additionally, I have included research findings on consumer participation in various types of projects. Finally, I have summarized the literature review, highlighting existing research gaps and the innovation of this study.Thank you.

4. The presentation of the method is very imprecise

Response :I have reviewed the parameter expressions, sentence coherence, and spelling errors in the methods section. I have also reorganized the language and improved some expressions that were incorrect or omitted.

5. Interpretation of results does not highlight important issues studied in this paper

Response :The article has added a discussion section, which analyzes the impact of the variations in the main parameters on the three parties and showcases the theme of the paper. Additionally, the conclusion section has included the main conclusions of the article.

6. The discussion section does not analyze the results of this paper, and provides enlightening thinking on related issues. the following literature should be helpful for your research：(1) Reduction pathways identification of Agricultural Water Pollution in Hubei Province, China. (2) Coordination of the Industrial-Ecological Economy in the Yangtze River Economic Belt, China.

Response :The article has added a discussion section, which analyzes the impact of the variations in the main parameters on the three parties and showcases the theme of the paper.Thank for your help.

7. Compared with the available literature, what are the theoretical contributions and application values of this study? It is suggested to enhance the corresponding discussions in the conclusion part.

Response :I have added two paragraphs in the conclusion section that discuss the impact of consumer participation on PPP project governance and the contribution of this study to the theory.

---

## [Decision Letter · Decision Letter 1]

19 Feb 2024

Decision Analysis of PPP Project’s Parties Based on Deep Consumer Participation

PONE-D-23-19775R1

Dear Dr. Liu,

We’re pleased to inform you that your manuscript has been judged scientifically suitable for publication and will be formally accepted for publication once it meets all outstanding technical requirements.

Kind regards,

Maurizio Fiaschetti

Academic Editor

PLOS ONE

Additional Editor Comments (optional):

Reviewers' comments:

Reviewer's Responses to Questions

**Comments to the Author**

1. If the authors have adequately addressed your comments raised in a previous round of review and you feel that this manuscript is now acceptable for publication, you may indicate that here to bypass the “Comments to the Author” section, enter your conflict of interest statement in the “Confidential to Editor” section, and submit your "Accept" recommendation.

Reviewer #1: All comments have been addressed

Reviewer #2: All comments have been addressed

2. Is the manuscript technically sound, and do the data support the conclusions?

Reviewer #1: Yes

Reviewer #2: Yes

3. Has the statistical analysis been performed appropriately and rigorously? 

Reviewer #1: Yes

Reviewer #2: Yes

4. Have the authors made all data underlying the findings in their manuscript fully available?

Reviewer #1: Yes

Reviewer #2: Yes

5. Is the manuscript presented in an intelligible fashion and written in standard English?

Reviewer #1: Yes

Reviewer #2: Yes

6. Review Comments to the Author

Reviewer #1: (No Response)

Reviewer #2: (No Response)

7. PLOS authors have the option to publish the peer review history of their article (what does this mean?). If published, this will include your full peer review and any attached files.

Reviewer #1: No

Reviewer #2: No

---

## [Editor Report · Acceptance letter]

2 Apr 2024

PONE-D-23-19775R1 

PLOS ONE

Dear Dr. Liu, 

I'm pleased to inform you that your manuscript has been deemed suitable for publication in PLOS ONE. Congratulations! Your manuscript is now being handed over to our production team.

Kind regards, 

on behalf of

Dr. Maurizio Fiaschetti 

Academic Editor

PLOS ONE